# SIRT1 Contributes as an Invasiveness Marker in Pituitary Adenoma

**DOI:** 10.3390/brainsci11121638

**Published:** 2021-12-11

**Authors:** Domantas Vaiciulis, Alvita Vilkeviciute, Greta Gedvilaite, Brigita Glebauskiene, Loresa Kriauciuniene, Rasa Liutkeviciene

**Affiliations:** 1LUHS, Medical Academy, Eiveniu 2, LT-50009 Kaunas, Lithuania; domantas.vaiciulis@gmail.com (D.V.); brigita.glebauskiene@lsmuni.lt (B.G.); 2Neuroscience Institute, LUHS, Medical Academy, Eiveniu 2, LT-50161 Kaunas, Lithuania; alvita.vilkeviciute@lsmuni.lt (A.V.); loresa.kriauciuniene@lsmuni.lt (L.K.); rasa.liutkeviciene@lsmuni.lt (R.L.)

**Keywords:** pituitary adenoma, SIRT1, rs3758391, rs3818292, rs7895833

## Abstract

The aim of the study was to find the association between SIRT1 concentration, *SIRT1* rs3758391, rs3818292, rs7895833 polymorphisms and clinical manifestations of pituitary adenoma (PA). The study included 108 patients with PA and 216 healthy individuals. Using commercial kits, DNA was extracted from peripheral blood leukocytes. To determine the PA and control group subjects genotypes was used real-time PCR method, for SIRT concentration measurement we used ELISA method. The statistical data analysis was completed using the “BM SPSS Statistics 20.0” software. Results: We performed statistical analysis of SNPs in the patient and healthy controls and patients’ subgroups and found statistically significant differences in rs7895833 genotype (A/A, A/G, G/G) distributions between the active PA and control groups (67.9%, 24.6%, 5.7% vs. 72.2%, 27.3%, 0.5%; *p* = 0.02) Also, the results showed that the rs7895833 G/G genotype is associated with about 13-fold increased odds of active PA development compared to the A/A (OR = 13.95% CI: 1.314–128.632; *p* = 0.028) and both A/A and A/G genotypes (OR = 12.9; 95% CI: 1.314–126.624; *p* = 0.028). There is ample evidence that SIRT1 in the pituitary and other target organs modifies the synthesis, secretion, and activity of hormones to trigger adaptive responses, thus we decided to include this in our study. When determining the serum concentration of SIRT1, we did not find a statistically significant difference between the PA group and the control group. SIRT1 serum level was statistically significantly higher in women with PA than in healthy control women (1.115 (3.748) vs. 136 (0.211); *p* = 0.008). To conclude—*SIRT1* rs7895833 G/G genotype is associated with about 13-fold increased odds of active PA development compared to the A/A and both A/A and A/G genotypes. SIRT1 serum levels are higher in women with PA than in healthy women.

## 1. Introduction

The pituitary gland is an endocrine gland located in a bone cavity, the sella turcica. The pituitary gland consists of two parts: the anterior lobe (adenohypophysis), which synthesizes and releases an adrenocorticotropic hormone, prolactin, growth hormone, thyrotropic hormone, follicle-stimulating hormone, and luteinizing hormone and the posterior lobe (neurohypophysis), which accumulates hormones secreted by the hypothalamus such as oxytocin and antidiuretic hormone [1,2,3]. Pituitary adenoma (PA) is the most common pituitary disorder, with a prevalence of 14.4% in pooled autopsy and about 22.5% in radiological studies [4]. The great attention in the PA pathogenesis is drawn to various genetic factors. The present study selected sirtuin 1 (SIRT1), associated with different types of cancers [5,6,7,8,9,10,11,12,13,14,15,16,17,18,19,20,21,22]. The *SIRT1* gene is found on the long shoulder of chromosome 10, 10q21.3, and is 33715 base pairs in size with nine exons encoding 747 amino acids. SIRT1 is a nicotinamide adenine dinucleotide-dependent deacetylase capable of deacetylating histonic and nonhistonic proteins and other transcription factors. It plays an essential role in cell death and survival, gene transcription, energy balance, and oxidative stress regulation through various processes interacting with p53Ku70 B cell lymphoma like protein 4 pathway, forkhead box O3, and others [7,8,23,24,25,26,27].

Some studies have shown an increase in the level of SIRT1 in various tumors such as hepatocellular carcinoma, breast, prostate, ovarian, gastric, and colon cancer, leukemia, melanoma, lymphoma, and glioblastoma [8,9,10,11,12,13,14,15,16,17]. Also, there is growing evidence that SIRT1 may play a role in the endocrine system [28,29]. Although evidence is accumulating for the importance of SIRT1 in the endocrine system. Therefore, we decided to investigate SIRT1 levels in patients with pituitary adenoma to know whether SIRT1 regulates metabolism in the brain and pituitary gland. 

*SIRT1* rs3758391 has been investigated in diffuse large B cell lymphoma, bladder, and breast cancer [6,17,18]. Our previous studies analyzed the *SIRT1* rs12778366 and rs3740051 polymorphisms, and we revealed that *SIRT1* rs3740051 was associated with PA recurrence and invasiveness and rs12778366 was associated with PA development [21,22].

Therefore, this study aimed to determine the association between SIRT1 concentration and gene polymorphisms and PA development.

## 2. Materials and Methods

Permission (No. P2-9/2003) to perform the study was obtained from the Ethics Committee for Biomedical Research. 

PA group and healthy controls group were included in our study. The PA group inclusion criteria were PA diagnosed and confirmed by magnetic resonance imaging (MRI); good general health; an informed consent to participate in the study; age 18 years and above; and the absence of other tumors.

Healthy controls group was composed according to the distribution of gender and age in the pituitary adenoma group. Therefore, the patients’ age didn’t differ statistically significantly (<0.05) between the pituitary adenoma and control groups.

Invasiveness, activeness and recurrence evaluation, and control group formation were described previously in our studies [21,22]. DNA extraction and genotyping were described previously in our studies, as well [21,22].

## 3. Statistical Analysis

In the present report, we calculated the deviation from HWE of the *SIRT1* (rs3818292, rs3758391, rs7895833) alleles. Pearson’s χ^2^ statistical test was used in both case and control subjects.

Data were analyzed by SPSS 20.3 statistical analysis software (IBM SPSS, Armonk, NY, USA). Frequencies of *SIRT1* (rs3818292, rs3758391, rs7895833) genotypes and alleles between case and control subjects were compared using the χ^2^ test in all groups. The nonparametric Mann-Whitney U test was used to compare non-normally distributed continuous data, and the Pearson’s χ^2^ test was used to compare categorical variables. Genotype-based Odds Ratio (OR) and 95% confidence intervals (Cis) were estimated using binomial logistic regression models regarding the impact of genotypes on PA development. Data are presented as absolute numbers with percentages in parentheses and median and interquartile range (IQR). *p*-value less than 0.05 was considered to indicate a statistically significant difference. Only statistically significant variables are presented in the tables.

## 4. Results

Our study enrolled 108 patients with PA and 216 healthy controls. The control group was formed according to the gender and age distribution in the PA group. The PA group was later accurately evaluated and divided into subgroups by PA’s hormonal activity, invasiveness, and recurrence. Demographic characteristics are described in Table 1.

### 4.1. Associations of SIRT1 rs3818292, rs3758391, rs7895833 with PA Development

Three SNPs at the *SIRT1* gene (rs3818292, rs3758391, and rs7895833) were determined for all study subjects. Hardy Weinberg equilibrium (HWE) analysis was performed to evaluate the genotype distributions in controls. The analysis showed that all three SNPs fulfilled the HWE criteria (*p* > 0.05). Frequencies of *SIRT1* rs3818292, rs3758391, rs7895833 genotypes and alleles were compared between the PA and control groups. Statistically significant differences were not observed (Table 2).

Binomial logistic regression analysis was performed to evaluate the impact of *SIRT1* (rs3818292, rs3758391, and rs7895833) gene polymorphisms on PA development; unfortunately, no statistically significant associations were found.

Further statistical analysis was performed to evaluate the *SIRT1* rs3818292, rs3758391, and rs7895833 associations with PA development in males and females separately, considering different PA pathogenesis in males and females. The analysis showed no statistically significant results.

### 4.2. Associations of SIRT1 rs3818292, rs3758391, rs7895833 with PA Development by PA’s Hormonal Activity, Invasiveness, and Recurrence

Hormonal activity, invasiveness, and recurrence were evaluated in all PA patients. We performed statistical analysis of SNPs in different subgroups of patients and found statistically significant differences in rs7895833 genotype A/A, A/G, and G/G distributions between the active PA and control groups (67.9 %, 24.6 %, and 5.7 % vs. 72.2 %, 27.3 % and 0.5 %; *p* = 0.02) (Table 3).

Binomial logistic regression showed that rs7895833 G/G genotype is associated with about 13-fold increased odds of active PA development under the codominant (OR = 13; 95% CI: 1.314–128.632; *p* = 0.028) and recessive (OR = 12.9; 95% CI: 1.314–126.624; *p* = 0.028) genetic models (Table 4).

Associations were not found between *SIRT1* rs7895833 and non-active PA development, recurrent or non-recurrent PA, and invasive or non-invasive PA development. Statistically significant associations were not found between *SIRT1* rs3818292 and rs3758391 and all PA subgroups.

### 4.3. SIRT1 Serum Levels in PA Patients and Controls

We evaluated SIRT1 serum levels in ten PA patients and 19 age and gender-matched control group subjects and determined that SIRT1 serum levels do not differ statistically significantly between the PA and control groups (0.810 (1.685) vs. 0.255 (0.293); *p* = 0.126) (Figure 1). We also compared the SIRT1 serum levels between different gender groups and found that women with PA have higher SIRT1 levels than control women (1.115 (3.748) vs. 136 (0.211); *p* = 0.008) (Figure 2). Any associations were found between the age of PA patients and SIRT1 concentrations.

Considering the associations between SNP genotypes and SIRT1 levels, we analyzed the SIRT1 levels in different genotype groups, but no statistically significant differences were found.

## 5. Discussion

In our study, we decided to evaluate the association between SIRT1 concentration, *SIRT1* rs3758391, rs3818292, rs7895833 polymorphisms, and clinical manifestations of pituitary adenoma.

Some studies have reported that several cell types, such as mouse astrocytes, microglia, oligodendrocytes, express SIRT1 proteins, which means that SIRT1 is expressed in CNS in animal models [30,31]. SIRT1 is also widely expressed in several human tissues, including the brain [32]. The impact of SIRT1 on cancer development, prognosis, or invasion has been studied in several studies. Cha et al. reported that SIRT1 expression was significantly related to poor prognosis in gastric carcinoma [13]. Noh et al. proved that SIRT1 expression was associated with lymph node metastasis and advanced tumor invasion [33]. It was also found that SIRT1 can have an impact on the metastasis of prostate cancer [34].

Several studies have analyzed SIRT1 in association with various types of brain tumors. A significant increase in the level of SIRT1 was revealed in glioblastoma [15]. Jing-Xin Ma et al. reported that SIRT1 expression was correlated with the formation and prognosis of human medulloblastomas [35].

Only a few studies are analyzing *SIRT1* gene polymorphisms in PA patients. In the study done by Glebauskiene et al., *SIRT1* (rs12778366) T/C genotype was less frequent in the PA group compared with the control group (0 vs. 17.5%, respectively; *p* < 0.001), the C/C genotype was found to be more frequent in the PA group compared with the healthy controls (18.9 vs. 2.5%, *p* < 0.001), which meant that SIRT1 could have had an impact on PA development [21]. Another study done by Liutkeviciene et al. revealed that *SIRT1* (rs3740051) might be associated with PA recurrence and invasiveness. The haplotype containing alleles C-A in rs12778366-rs3740051 was found to be associated with increased odds of PA development as well [22]. Our study found *SIRT1* rs7895833 G/G genotype is associated with about 13-fold increased odds of active PA development compared to the A/A and both A/A and A/G genotypes. This study evaluated the serum level of SIRT1 in PA patients and its impact on PA characteristics. We compared the SIRT1 serum levels between different gender groups and found that women with PA have higher SIRT1 levels than control women (1.115 (3.748) vs. 136 (0.211); *p* = 0.008). Prolactinomas, ACTH releasing adenomas, and TSH releasing adenomas are more common in females, while endocrine-inactive and GH releasing adenomas more common in males [36]. It is known, that SIRT1 works as an important biomarker in the molecular mechanisms of TSH pathway regulation [28]. This may be the reason why we found statistically significant differences between females and males. Therefore, no other studies have analyzed SIRT1 serum levels in patients with PA. We found just two studies analyzing SIRT1 serum levels in other types of cancer. Shaker et al. analyzed serum SIRT1 levels in patients with colorectal cancer. It was found that the SIRT1 level was higher in patients with colorectal cancer than in healthy subjects (737.3 pg/mL vs. 443.80 pg/mL; *p* = 0.001) [20]. In their study, Rizk et al. found that SIRT1 levels in patients with breast cancer were twice as high as in healthy controls (*p* < 0.0001) [17].

## 6. Conclusions

In conclusion, *SIRT1* rs7895833 G/G genotype is associated with about 13-fold increased odds of active PA development compared to the A/A and both A/A and A/G genotypes. SIRT1 serum levels are higher in women with PA than in healthy women. Of course, further studies are required to evaluate gender role in PA development and association with SIRT1. 

## Figures and Tables

**Figure 1 brainsci-11-01638-f001:**
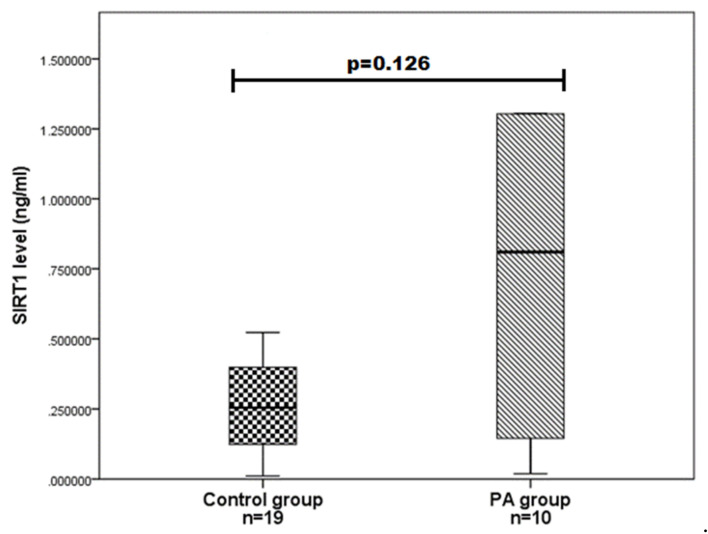
SIRT1 serum levels in PA and control groups. SIRT1 levels (ng/mL) in the PA patient serum versus controls are presented as box-and-whisker plots with the median and IQR. Mann–Whitney U test was used to assess the differences in SIRT1 concentration between PA patients and the control group.

**Figure 2 brainsci-11-01638-f002:**
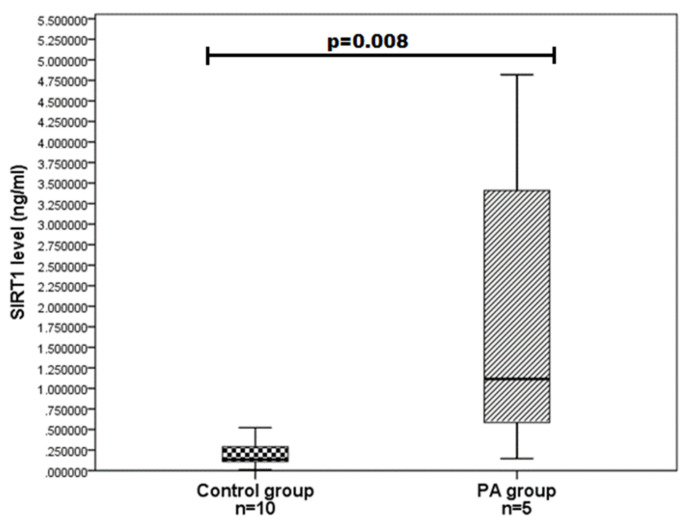
SIRT1 serum levels in PA and control group women. SIRT1 levels (ng/mL) in women’s serum with PA versus controls are presented as box-and-whisker plots with the median and IQR. Mann–Whitney U test was used to assess the differences in SIRT1 concentration between women with PA and control women.

**Table 1 brainsci-11-01638-t001:** Patients with pituitary adenoma (PA) and the control group subjects demographic characteristics.

Characteristic	Group		*p*-Value
	PA, *n* (%) (*n* = 108)	Control, *n* (%) (*n* = 216)	
Males	43 (39.8)	83 (38.4)	1.0
Females	65 (60.2)	133 (61.6)	1.0
Age, median, (min.–max.)	54 (18–84)	56 (19–94)	0.792
Hormonal activity * Active Non-active	53 (53) 47 (47)	-	-
Recurrence * Non-recurrent With recurrence	78 (78) 22 (22)	-	-
Invasiveness * Invasive Non-invasive	65 (63.1) 38 (36.9)	-	-

* Hormonal activity, recurrence, and invasiveness were evaluated in 100 PA patients; PA—pituitary adenoma; *p*-value—significance level (statistically significant when *p* < 0.05).

**Table 2 brainsci-11-01638-t002:** Frequencies of rs3818292, rs3758391, and rs7895833 genotypes and alleles in the PA and control groups.

SNP	Genotype/Allele	Frequency			*p*-Value
		Control Group, *n* (%)	HWE *p* Value	PA, *n* (%)	
rs3818292	Genotype				
	A/A	190 (88)	0.366	91 (88.3)	0.865
	A/G	25 (11.6)		11 (10.7)	
	G/G	1 (0.4)		1 (1)	
	All	216 (100)		103 (100)	
	Allele				
	A	405 (93.8)		193 (93.7)	0.976
	G	27 (6.2)		13 (6.3)	
rs3758391	Genotype				
	C/C	127 (58.8)	0.941	52 (50.5)	0.240
	C/T	77 (35.6)		47 (45.6)	
	T/T	12 (5.6)		4 (3.9)	
	All	216 (100)		103 (100)	
	Allele				
	C	331 (76.6)		151 (73.3)	0.362
	T	101 (23.4)		55 (26.7)	
rs7895833	Genotype				
	A/A	156 (72.2)	0.064	75 (72.8)	0.178
	A/G	59 (27.3)		25 (24.3)	
	G/G	1 (0.5)		3 (2.9)	
	All	216 (100)		103 (100)	
	Allele				
	A	371 (85.9)		175 (84.9)	0.755
	G	61 (14.1)		31 (15.1)	

PA—pituitary adenoma; *p*-value—significance level (statistically significant when *p* < 0.05); HWE *p*-value—Hardy-Weinberg significance level (statistically significant when *p* < 0.05).

**Table 3 brainsci-11-01638-t003:** Frequencies of *SIRT1* rs3818292, rs3758391, and rs7895833 genotypes and alleles in the active and non-active PA and control groups.

SNP	Genotype/Allele	Frequency					
		Non-Active PA, *n* (%) *n* = 47	Control Group, *n* (%) *n* = 216	*p*-Value	Active PA, *n* (%) *n* = 53	Control Group, *n* (%) *n* = 216	*p*-Value
rs3818292	Genotype						
	A/A	43 (91.5)	190 (88)	0.833	45 (84.9)	190 (88)	0.449
	A/G	4 (8.5)	25 (10.6)		7 (13.2)	25 (10.6)	
	G/G	0 (0)	1 (0.4)		1 (1.9)	1 (0.4)	
	Allele						
	A	90 (95.7)	405 (93.8)	0.457	97 (91.5)	405 (93.8)	0.408
	G	4 (4.3)	27 (6.2)		9 (8.5)	27 (6.2)	
rs3758391	Genotype						
	C/C	22 (46.8)	127 (58.8)	0.115	28 (52.8)	127 (58.8)	0.693
	C/T	24 (51.1)	77 (35.6)		22 (41.5)	77 (35.6)	
	T/T	1 (2.1)	12 (5.6)		3 (5.7)	12 (5.6)	
	Allele						
	C	68 (72.3)	331 (76.6)	0.379	78 (73.6)	331 (76.6)	0.512
	T	26 (27.7)	101 (23.4)		28 (26.4)	101 (23.4)	
rs7895833	Genotype						
	A/A	36 (72)	156 (72.2)	0.767	36 (67.9)	156 (72.2)	**0.02**
	A/G	11 (27.5)	59 (27.3)		14 (26.4)	59 (27.3)	
	G/G	0 (0)	1 (0.5)		3 (5.7)	1 (0.5)	
	Allele						
	A	83 (88.3)	371 (85.9)	0.536	86 (81.1)	371 (85.9)	0.221
	G	11 (11.7)	61 (14.1)		20 (18.9)	61 (14.1)	

PA—pituitary adenoma; *p*-value—significance level (statistically significant when *p* < 0.05). *p*-values marked in bold are statistically significant.

**Table 4 brainsci-11-01638-t004:** Associations of rs7895833 with active PA development.

rs7895833				
Genetic Model	Genotype/Allele	OR (95 % CI)	*p* Value	AIK
Codominant	A/A	1		265.173
	A/G	1.028 (0.518–2.042)	0.937	
	G/G	13 (1.314–128.632)	**0.028**	
Dominant	A/A	1		268.605
	A/G + G/G	1.228 (0.642–2.350)	0.536	
Recessive	A/A + A/G	1		263,179
	G/G	12.9 (1.314–126.624)	**0.028**	
Overdominant	A/A + G/G	1		268.966
	A/G	0.955 (0.484–1.886)	0.895	
Additive	G	1.451 (0.812–2.592)	0.209	267.452

*p*-value—significance level (statistically significant when *p* < 0.05); OR—odds ratio; AIK—Akaike information criterion. *p*-values marked in bold are statistically significant.

## Data Availability

Data will be provided in case a request is made by editors, reviewers, or scientists.

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
