# Peer review of "SIRT1 Contributes as an Invasiveness Marker in Pituitary Adenoma"

_brainsci, 2021, doi:10.3390/brainsci11121638_

Round 1

Reviewer 1 Report

In this study, Domantas Vaiciulis and colleagues have investigated the association between SIRT1 concentration, SIRT1 rs3758391, rs3818292, rs7895833 polymorphisms and clinical manifestations of pituitary adenoma. They performed statistical analysis of SNPs in the patient and healthy controls and patients' subgroups and found statistically significant differences in rs7895833 genotype (A/A, A/G,G/G) distributions between the active PA and control groups (67.9 %, 24.6 %, 5.7 % vs. 72.2 %, 27.3%, 0.5 %; p=0.02) Additionally, the results showed that the rs7895833 G/G genotype is associated with about 13-fold increased odds of active PA development compared to the A/A (OR=13; 95% CI: 1.314 – 128.632; p=0.028) and both A/A and A/G genotypes (OR=12.9; 95% CI: 1.314 – 126.624; p=0.028). SIRT1 serum level were statistically significantly higher in women with PA than in healthy control women (1.115 (3.748) vs. 136 (0.211); p=0.008). Suggesting, SIRT1 rs7895833 G/G genotype is associated with about 13-fold increased odds of active PA development compared to the A/A and both A/A and A/G genotypes. SIRT1serum levels were higher in women with PA than in healthy women.

All-in-all this is an interesting study demonstrating that SIRT1 rs7895833 G/G genotype is associated PA development compared to the A/A and both A/A and A/G genotypes. The experiments appear to be conducted in a systematic way, and data obtained were analyzed properly. The set of analytical methods used are robust, the research theme is relevant, and the results are interesting and intriguing. The manuscript is written well and discussed properly. Following concerns need attention.

  1. The introduction section lacks information about SIRT1 and the importance of using it as a biomarker for PA.

  1. Were there any differences in SIRT1 on PA patients based on age besides the increased found in PA vs. healthy females?

  1. The conclusion mentions further studies are required to evaluate gender role in PA development and association but lacks to explain why. Why are there such mark differences in males compared to females when assessing SIRT1?

Author Response

Dear Editor and Reviewers,

We kindly appreciate the revision of our manuscript. We have highlighted the changes we made in the manuscript by using the track changes mode in MS Word. Hope that the revised manuscript will be acceptable for publication in your journal. Enclosed please also find our point-by-point response to the comments raised by the reviewers.

Reviewer 1:

Comments and Suggestions for Authors

In this study, Domantas Vaiciulis and colleagues have investigated the association between SIRT1 concentration, SIRT1 rs3758391, rs3818292, rs7895833 polymorphisms and clinical manifestations of pituitary adenoma. They performed statistical analysis of SNPs in the patient and healthy controls and patients' subgroups and found statistically significant differences in rs7895833 genotype (A/A, A/G, G/G) distributions between the active PA and control groups (67.9 %, 24.6 %, 5.7 % vs. 72.2 %, 27.3%, 0.5 %; p=0.02) Additionally, the results showed that the rs7895833 G/G genotype is associated with about 13-fold increased odds of active PA development compared to the A/A (OR=13; 95% CI: 1.314 – 128.632; p=0.028) and both A/A and A/G genotypes (OR=12.9; 95% CI: 1.314 – 126.624; p=0.028). SIRT1 serum levels were statistically significantly higher in women with PA than in healthy control women (1.115 (3.748) vs. 136 (0.211); p=0.008). Suggesting, SIRT1 rs7895833 G/G genotype is associated with about 13-fold increased odds of active PA development compared to the A/A and both A/A and A/G genotypes. SIRT1serum levels were higher in women with PA than in healthy women.

 All-in-all this is an interesting study demonstrating that SIRT1 rs7895833 G/G genotype is associated with PA development compared to the A/A and both A/A and A/G genotypes. The experiments appear to be conducted in a systematic way, and the data obtained were analyzed properly. The set of analytical methods used are robust, the research theme is relevant, and the results are interesting and intriguing. The manuscript is written well and discussed properly. Following concerns need attention.

  1. The introduction section lacks information about SIRT1 and the importance of using it as a biomarker for PA.

It was added.

  1. Were there any differences in SIRT1 on PA patients based on age besides the increased found in PA vs. healthy females?

PA patients were divided into young and old based on the median age (median age was 56 years). SIRT1 serum concentrations were compared between these two groups, but it did not show any statistically significant differences between younger and older PA patients. Also, the SIRT1 serum concentration did not correlate with the age in overal PA group.

  1. The conclusion mentions further studies are required to evaluate gender role in PA development and association but lacks to explain why. Why are there such mark differences in males compared to females when assessing SIRT1?

Prolactinomas, ACTH releasing adenomas, and TSH releasing adenomas are more common in females, while endocrine-inactive and GH releasing adenomas are more common in males (Mindermann T, Wilson CB. Age-related and gender-related occurrence of pituitary adenomas. Clin Endocrinol (Oxf). 1994 Sep;41(3):359-64). It is known, that SIRT1 works as an important biomarker in the molecular mechanisms of TSH pathway regulation (Akieda-Asai, S., Zaima, N., Ikegami, K., Kahyo, T., Yao, et. al. SIRT1 Regulates Thyroid-Stimulating Hormone Release by Enhancing PIP5Kgamma Activity through Deacetylation of Specific Lysine Residues in Mammals. PloS one. 2010. 5(7), e11755). This may be the reason why we found statistically significant differences between females and males.

Reviewer 2 Report

Major concerns:

-The size of the population: healthy group control is two times bigger than patient group. I would adjust the values to compare the same.

- Could you also relate the SirT1 levels with the invasiveness and recurrence of the tumor?

Minor changes:

-Report in the abstract why you want to study Sirt1 levels in PA. 

Author Response

Dear Editor and Reviewers,

We kindly appreciate the revision of our manuscript. We have highlighted the changes we made in the manuscript by using the track changes mode in MS Word. Hope that the revised manuscript will be acceptable for publication in your journal. Enclosed please also find our point-by-point response to the comments raised by the reviewers.

Major concerns:

-The size of the population: healthy group control is two times bigger than patient group. I would adjust the values to compare the same.

We enroled two controls for every case to add power to our study.

Lewallen S, Courtright P. Epidemiology in practice: case-control studies. Community Eye Health. 1998;11(28):57-58.

- Could you also relate the SirT1 levels with the invasiveness and recurrence of the tumor?

We compared SIRT1 serum concentrations between patients with invasive PA and noninvasive PA, but no statistically significant differences were found. Furthermore, we could not compare SIRT1 serum concentrations between PA with recurrence and without recurrence because of too small number of patients in PA without recurrence group. Moreover, for such subgroup analysis the number of patients should be increased for the SIRT1 serum concentration evaluation.

Minor changes:

-Report in the abstract why you want to study Sirt1 levels in PA. 

Information added.
